# Effects of gestational age on blood cortisol and prolactin levels during pregnancy in malaria endemic area

**Francois Kiemde** [1] *, **Hermann Sorgho**[1], **Serge Henri Zango**[1], **Gnohion Fabrice Some**[1], **Toussaint Rouamba**[1], **Ousmane Traore** [1], **Berenger Kabore**[1], **Hamtandi Magloire Natama**[1], **Yeri Esther Hien**[2], **Innocent Valea**[1], **Henk Schallig**[3], **Halidou Tinto** [1]

**1** Institut de Recherche en Sciences de la Santé–Clinical Research Unit of Nanoro (IRSS-CRUN), Nanoro, Burkina Faso, **2** Unité de Recherche et de Formation en Sciences de la Vie et de la Terre (URF-SVT), Université Joseph Ki-Zerbo Ouaga 1, Ouagadougou, Burkina Faso, **3** Department of Medical Microbiology and Infection Prevention, Laboratory for Experimental Parasitology, Amsterdam Institute for Infection and Immunity, Amsterdam University Medical Centres, Academic Medical Centre at the University of Amsterdam, Amsterdam, The Netherlands

* kiemdefrancois@yahoo.fr

**Data Availability Statement:** All relevant data are within the paper and its Supporting Information files.

## Abstract

### Background

The hormonal shift occurring in pregnant women is crucial for the outcome of pregnancy. We conducted a study in pregnant women living in a malaria endemic area to determine the potential effect of gestational age on the modulation of the endocrine system by cortisol and prolactin production during pregnancy.

### Methods

Primigravidae and multigravidae with a gestational age between 16–20 weeks were included in the study and followed up to delivery and 6–7 weeks thereafter. Venous blood was collected at scheduled visit: Visit 1 (V1; 16–20 weeks of amenorrhea), Visit 2 (V2; 28 ±1 weeks of pregnancy), Visit 3 (V3; 32 ±1 weeks of pregnancy), Visit4 (V4; delivery) and Visit5 (V5; 6–7 weeks after delivery). In addition, a cord blood sample was also collected during labour at delivery. Nulliparous and primiparous/multiparous non-pregnant women were enrolled in the control group. Cortisol and prolactin plasma concentrations were measured using ichroma II and i-chamber apparatus. Light microscopy was used to detect *Plasmodium falciparum* infections. A linear mixed-effects regression (LMER) model was used to assess the association between the variation of cortisol titres and prolactin levels during the pregnancy and the post-partum.

### Results

Results showed that cortisol and prolactin levels in the peripheral blood were globally up-regulated during pregnancy. Concentrations of cortisol during follow-up was significantly higher in primigravidae than in multigravidae during the whole pregnancy (p<0.024). Moreover, the level of prolactin which was higher before delivery in primigravidae reversed at

**Funding:** International Society for Infectious Diseases (ISID), Small Grant 2014, "Determination of cytotoxicity of NK cells against erythrocytes infected by Plasmodium Falciparum during pregnancy". the funder had no role in the study design, data collection and analysis, decision to publish, or preparation of the manuscript.

**Competing interests:** The authors have declared that no competing interests exist.

**Abbreviations:** ANC, antenatal consultation; CRUN, Clinical Research Unit of Nanoro; EDTA, Ethylene diamine tetra acetic acid; IPT, immunologic pregnancy test; LMER, linear mixed-effects regression; *NK, Natural Killer*; SSA, sub-Saharan Africa.

delivery and postpartum visit, but the difference was not statistically significant during the follow-up (V1 to V5) (p = 0.60). The cortisol level in peripheral blood at delivery was higher than that in the cord blood, and conversely for prolactin. Cortisol and prolactin levels decreased after delivery, though the level of prolactin was still higher than that at enrolment. An increase of one unit of prolactin was associated with the decrease of the average concentration of cortisol by 0.04 ng/ml (p = 0.009). However, when cortisol increases with one unit, the average concentration of prolactin decreases by 1.16 ng/ml (p = 0.013).

## Conclusion

These results showed that the up-regulation effects of cortisol and prolactin are related to gestational age. A The downward regulation effect that both hormones have on each other during the pregnancy when each increase to 1 unit (1.0 ng/ml) was also reported.

## 1. Background

The maternal immune system plays an important role in the protection of the mother against environmental pathogens and to prevent the developing foetus against any damage [1–8]. The immunological variation of the mother during pregnancy is crucial in the outcome of pregnancy. However, in sub-Saharan Africa (SSA), some infections during pregnancy, like malaria, lead to complications. These complications lead to important burden for both mothers and their new-borns and thus represent a real public health concern, in particular among women experiencing their first or second pregnancy [9].

For many years, the focus on the concept of immunology of pregnancy as an organ transplant has complexified research in the field of immunity development during pregnancy and delayed the development of new pathways with clinical implications that could help to answer the existing gaps and other relevant questions [10]. Among these gaps and relevant question, the "suppression of (some) immune responses" concept ranks among those that have been accepted by all [11]. This concept was supported by the idea that the maternal immune response is biased towards humoral immunity [12], which could explain the suppression of immune retaliation against the developing foetus. Studies consider immunosuppression as a state in which immune system does not work as well it should. This mechanism provided to mother an effective protection from harmful pathogens while allowing tolerance to foetal antigens [13, 14]. However, several studies have shown that the development of the placenta changes and adapts the immune system to pregnancy without suppressing it [15]. This testifies the capacity of pregnant women to develop immune responses when the mother and/or the foetus are threatened [1, 2]. However, this association between the foetus and the immune system of the pregnant woman supports the hypothesis that the (hormonal) changes made during pregnancy and the adaptation of the immune system to pregnancy could depend on gestation (number of pregnancies).

Pregnancy is marked by a complex interplay of hormonal shifts, with cortisol and prolactin being two of the pivotal hormones regulating various maternal and foetal processes [16]. Cortisol, a glucocorticoid produced by the adrenal cortex, plays a crucial role in metabolic functions, stress responses, and the maturation of foetal organs [17]. Prolactin, secreted by the anterior pituitary, not only prepares the body for breastfeeding but also modulates the maternal immune system and foeto-maternal interface [18]. The hormonal changes that occur

during pregnancy contribute significantly to immunological changes [19, 20]. This is the case of stress responses on immune system during pregnancy. Indeed, the immune stress can be bidirectional by enhancing or suppress immune function [21, 22]. Cortisol and prolactin are among the hormones whose concentrations vary during pregnancy [23, 24]. The relationship between these hormones and the immune system has been demonstrated [25]. Cortisol acts as an immunosuppressive hormone [26, 27], whereas prolactin functions as an immunostimulatory hormone [28–30]. Although several studies reported the association between cortisol and prolactin levels with malaria in pregnancy [31–33], very few studies have investigated the evolution of cortisol and prolactin concentrations according to the stage and age of gestation [34]. In such context, more studies are needed to better understand the dynamic of cortisol and prolactin in peripheral blood during pregnancy and in umbilical cord blood at delivery. This could contribute to adapt prevention strategies against infectious diseases, especially in malaria endemic settings. The present study was designed to investigate the potential effect of gestation in modulating cortisol and prolactin levels in peripheral blood samples during pregnancy and in cord blood at delivery in a malaria endemic area.

## 2. Methods

### Study site

The study was conducted in the health district of Nanoro, located in the Center-West region of Burkina Faso. Nanoro is situated approximately 85 km away from Ouagadougou, the capital city. Malaria, caused by *Plasmodium falciparum*, remains the first cause of medical consultation in pregnant women in this region and predominately occurs during the rainy season July-November which overlap with the peak of malaria transmission.

Participants were enrolled in three (03) peripheral health facilities of the district (Nanoro, Nazoanga and Seguedin) where biological samples were collected and transported to the central laboratory of the Clinical Research Unit of Nanoro (CRUN) for processing and storage.

### Study design

This was a cohort study (2 cohorts), comprising prolactin and cortisol variation during pregnancy in primigravidae and multigravidae women. Non pregnant nulliparous and primiparous/multiparous were also enrolled in this study as control group and sampling once during the study for cortisol and prolactin titter. Pregnant women (primigravidae and multigravidae) attending the antenatal consultation (ANC) at one of the participating health facilities with fundal height less than 20 cm were invited to participate to the study. Written informed consent was obtained from each participant prior to the samples and data collection. Gestational age was determined by ultrasound and only participants with a pregnancy aged between 16 to 20 weeks were enrolled and followed-up until delivery and 6–7 weeks thereafter. During the follow-up, venous blood was collected in EDTA (Ethylene diamine tetra acetic acid) tube at the following timepoints:16th-20th weeks of pregnancy (V1); 28th±1 week of pregnancy (V2), 32th±1 week of pregnancy (V3), delivery (V4) and 6th-7th weeks after delivery (V5). Regarding the rationale behind these time points, the choice of 16th to 20th weeks of pregnancy is related to the fact that pregnant women consult to their first antenatal consultation at this period, due to socio-economic factors. This also represent the end of organogenesis. The choices of 28th±1 weeks of pregnancy is related to the maturation of the different organs of the foetus and 32th±1 to the pre-delivery. A cord blood sample was also collected at delivery. Pregnant women were encouraged to deliver at the participating health facilities to receive adequate care and facilitate the collection of blood samples (venous blood during the labour and umbilical cord blood at delivery).

For the control groups, participants were recruitment within the catchment area of the recruitment health facilities listed above. In this group, participants included were non-pregnant nulliparous and both primiparous/ multiparous individuals. The number of nulliparous and primiparous/ multiparous to be recruited by health facility catchment area were paired respectively to the number of primigravidae and multigravidae women recruited in this health facility catchment area. For each participant enrolled in the control group, venous blood was drawn once into EDTA tubes. Written informed consent was also signed by each non-pregnant women before data collection. No follow-up visit was performed for this group. To ensure that control participants were not pregnant, an immunologic pregnancy test (IPT) was performed and only women with negative IPT were enrolled in the control arm. Those for whom the IPT turned positive were referred to the maternity for appropriate care. Enrolment was carried out from July to November 2018 for pregnant women and from April to July 2019 for non-pregnant women.

## Operational definition

- Primigravidae women refer to women who were pregnant for the first time;

- Multigravidae women refer to women who were pregnant more than one time;

- Nulliparous women refer to women who were never give birth;

- Primiparous/multiparous women refer to women who already gave birth at least one time.

## Ethical consideration

This study was approved by the institutional ethics committee of Centre Muraz of Bobo Dioulasso, Burkina Faso (N/Réf 000013 of 24 March 2015).

## Laboratory procedure

**Blood samples collection and transport.**   The blood samples collected in this study were consisted of maternal venous blood collected at each scheduled visit including delivery (during labour) and umbilical cord blood. The blood samples were collected in EDTA tube. Only 3 ml of blood (venous or umbilical cord) was collected. The blood samples were immediately transported to the clinical laboratory of CRUN under cool condition (+4°C). Pregnant women were encouraged to give birth at the health center in order to receive more adequate health care and to facilitate the sampling of venous blood during labour and umbilical cord blood after delivery. The umbilical cord blood was collected after clamping the umbilical cord at 2.5 cm above its foetal insertion.

For non-pregnant women, a single blood (venous) sample was taken after a negative IPT.

**Biochemical test of cortisol and prolactin.**   Plasma samples were obtained by centrifuged the EDTA tubes at 2,000G for 10 minutes to separate the pellet from the plasma. All plasma were collected in labelled sterile tubes and then stored at -20°C until its used for biochemical testing within the 30 days.

Plasma concentrations of cortisol and prolactin were determined by using the test device ichroma™ cortisol for the quantification of cortisol and ichroma™ prolactin for the quantification of prolactin on the devices i-chamber (Weldon Biotech India Pvt. Ltd. India) and ichroma II (Boditech Med Inc. Korea) according to the procedure described by the manufacturer. For the quantification of the cortisol, 30µl of plasma were transferred to the tube containing the detection buffer provided by the manufacturer and mixed 10 times manually by

pipetting. A volume of 75μl of the mixture were placed in the well of the ichroma™ cortisol cartridge and immediately incubated at 25˚C for 10 minutes in the i-chamber. After incubation, the cartridge was removed and read immediately in the ichroma II. The analyzer automatically displayed the test result on the display screen, printed and validated. The procedure of quantification of prolactin was similar to cortisol, except the volume of plasma pipetted (75μl instead of 30μl).

The threshold for prolactin quantification with ichroma II is 100ng/ml. In case of concentration over 100ng/ml, ¼ and $\frac{1}{8}$ dilution of the plasma was done with saline phosphate buffer as recommended by the manufacturer. The exact value of the prolactin was then obtained by multiplying the measured value by the dilution factor.

**Malaria microscopy.** Malaria diagnosis by microscopy was performed by expert microscopists at CRUN central laboratory. Thick and thin blood smears were prepared (in duplicate) from venous blood collected in EDTA tubes. Thin films were fixed with methanol and blood slides were stained with 10% Giemsa solution for identification and quantification of asexual *Plasmodium falciparum* and other *Plasmodium* species. Parasite density was determined in thick blood smear by counting the number of asexual parasites per 200 leucocytes and estimated per μl of blood, based on an assumed white blood cells of 8,000 per μl by light microscopy. Thick blood smear was considered negative when the examination of 100 thick film fields did not reveal the presence of any asexual parasites. Each blood slide was read by two independent readers, and in case of discrepancy (positive *versus* negative, difference in *Plasmodium* species, difference in parasite density >Log10 or ratio>2 in case of parasite density ≤400/μl and >400/μl respectively) the blood slide was read again by a third independent reader whose conclusion was decisive. Positive microscopy results were recorded as the geometric means of the results from both readers results or the geometric means of the two geometrically closest reading in case of third reading. These results were expressed as asexual parasites per microliter by using the patient's white blood cell (WBC) count. A selection of slides (5%) was re-read by an independent expert microscopist for quality control.

## Sample size and data analysis

This study is part of a study that aimed to determine the cytotoxicity of *Natural Killer* (*NK*) cells against erythrocytes infected by *Plasmodium falciparum*. The sample size was determined in the framework of the *NK* cells cytotoxicity study. Briefly, the samples size was determined by considering the cytotoxicity of *NK* cells. Considering the results reported by Mavoungou et *al.* 2003, we estimated a standard deviation (SD) = 0.29, the effect size (EF) = 0.51, α = 0.05, number of groups = 4, using on F test with power = 0.8, our sample size was 48. By considering 10% of non-response (miss of follow-up), the sample size was estimated at 48 + 4 = 52 participants, meaning 13 women for each cohort (primigravidae and multigravidae) and 13 women per control group (nulliparous and primiparus/multiparous).

Data were collected on paper case report forms (CRF) and subsequently entered using REDCap (Research Electronic Data Capture) software.

The data analysis was done by using R software (R Foundation for Statistical Computing, Vienna, Austria). Description of qualitative and quantitative variables were performed by using proportion and mean ± SD (or median with IQR where appropriate), respectively. For the comparison cortisol and prolactin concentrations between gravity, the Wilcoxon rank-sum test was used. A repeated measure ANOVA test was used to compare the log value of cortisol and prolactin between the visits.

To assess the association between the overall change of cortisol and prolactin, we used a linear mixed-effects regression (LMER) model, which accounts for both fixed-effect factors and

**Table 1. Characteristic of study population.**

| Parameters | Pregnant women | | Non-pregnant women | |
|---|---|---|---|---|
| | Primigravidae | Multigravidae | Nulliparous | Primiparous/ Multiparous |
| Age (years) (SD) | 18.9 (2,5) | 25.8 (5.4) | 18.9 (2.0) | 34.1 (4.7) |
| Gestational age (weeks) [IQR] | 17.6 [16.4–18.4] | 17.6 [16.9–18.0] | - | - |
| Malaria diagnosis (microscopy), n (%) | | | 3 (20.0) | 0 (0) |
| V1 (16–20 weeks) | 10 (76.9) | 5 (38.5) | - | - |
| V2 (28±1 weeks) | 1 (7.7) | 1 (7.7) | - | - |
| V3 (32±1 weeks) | 2 (15.4) | 0 (0) | - | - |
| V4 (delivery) | 0 (0) | 2* (15.4) | - | - |
| V5 (6–7 weeks after delivery) | 1 (7.7) | 1 (7.7) | - | - |

*1 microscopy was positive for cord blood.

Table 1 summarize the characteristic of study population. Globally malaria was mostly diagnosed at enrolment and could be due to the fact that most of them were at their first antenatal visit and were not under IPTp.

random-effect factor (subject and visit). The fixed-effect factors included in the LMR was gravidity, parasitaemia and cortisol (if the outcome variable is prolactin) or prolactin (if the outcome variable is cortisol). A p value <0.05 was considered as statistically significant.

## 3. Results

### Characteristic of the study population

The basic characteristics of study population are presented on Table 1. A total of 56 pregnant women with fundal height less than 20 cm were screened and 50% (28/56) with gestational age between 16[th]-20[th] weeks (13 primigravidae and 15 multigravidae) were enrolled for the follow-up. The per-protocol (PP) population included 24 pregnant women (12 primigravidae and 12 multigravidae): 2 multigravidaes were loss of follow-up, 1 multigravidaes has delivered outside the study area and 1 primigavidae was dead. In total, 23 women were delivered vaginally and a primigravidae by caesarean. For non-pregnant women 27 (15 nulliparous and 12 primiparous/ multiparous) were enrolled.

In the pregnant women group, the mean age was 18.9 years (Standard deviations (SD): 2.5) in primigravidae cohort *versus* 25.8 years (SD: 5.4) in multigravidae cohort. Gestational age was 17.6 weeks (interquartile range (IQR): 16.4–18.4) for primigravidae and 17.6 weeks (IQR: 16.9–18.0) for multigravidae. In the non-pregnant women group, the mean age was 18.9 years (SD: 2.0) and 34.1 years (SD: 4.7) for nulliparous and primiparous/multiparous, respectively.

### Variation of cortisol during pregnancy

A gradual increase in cortisol concentrations from the enrolment to delivery was observed (primigravidae 99.2 ng/ml [86.4–130.1]; multigravidae 94.4 ng/ml [85.9–134.5]), with a decrease between V2 (primigravidae 127.7 ng/ml [111.7–134.4]; multigravidae 116.4 ng/ml [98.8–124.2]) and V3 (primigravidae 108.2 ng/ml [97.6–124.5]; multigravidae 107.6 ng/ml [100.2–119.1]), and the highest concentrations measured at delivery (primigravidae 173.0.2ng/ml [160.0–194.0]; multigravidae 128.3 ng/ml [118.8–174.7]). These concentrations however decreased at 6–7 weeks postpartum reaching 79.3 ng/ml [50.4–96.5] for primigravidae and 59.2 ng/ml [47.1–88.2] for multigravidae. Primigravidae showed a higher increase of cortisol than multigravidae during pregnancy with statistically significant different (p = 0.033). This increase of cortisol concentration across visits was statistically significant from 32[th]±1 weeks of

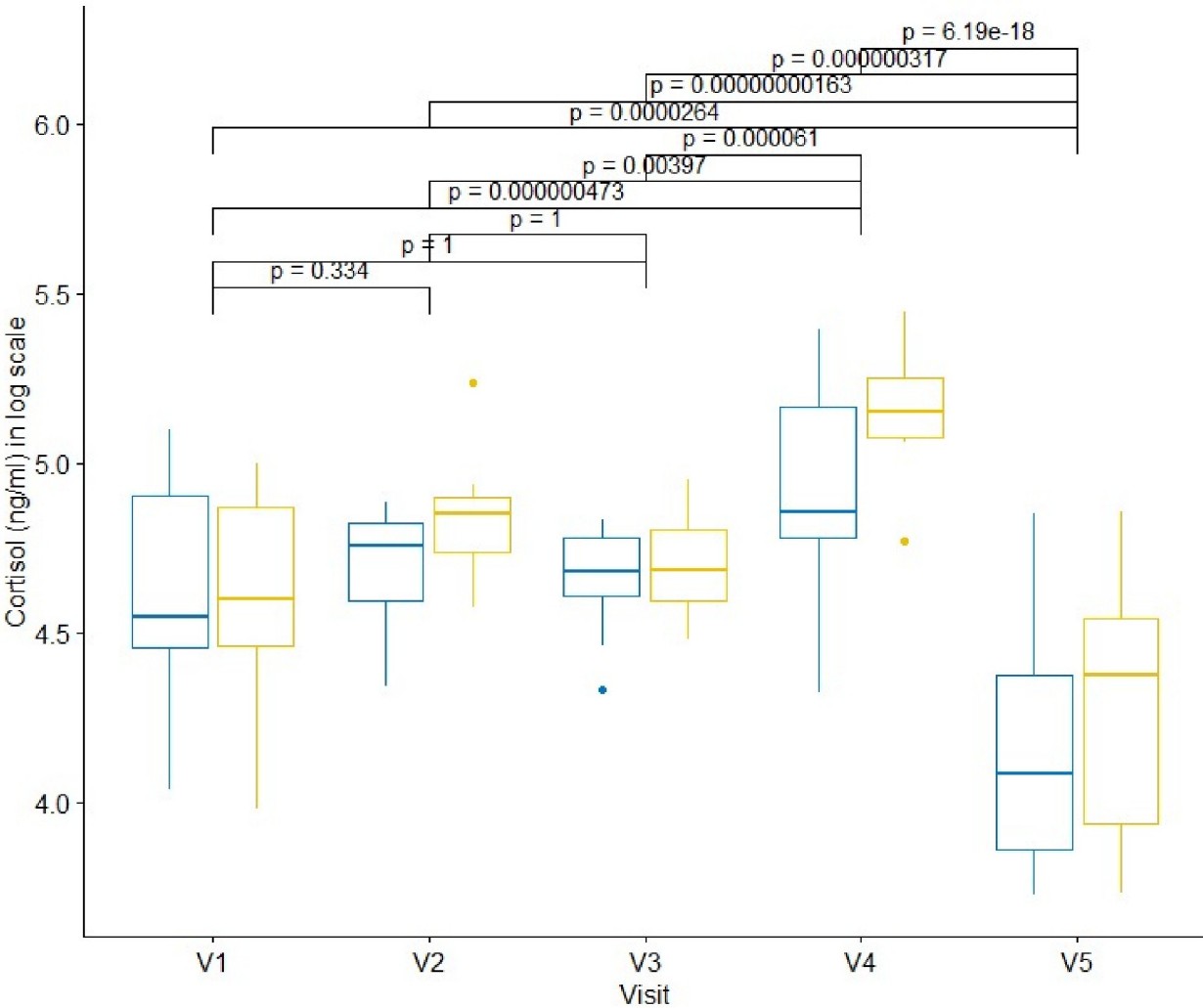

**Fig 1. Dynamics of cortisol levels measured in pregnant women during pregnancy determined at different ANC visits.** Legend: NS: not significant; V: visit. Code: *: <0.05; **: <0.01; ***: <0.001; ****: <0.0001. In each case, 30μl of plasma was used to determine for cortisol level in blood (ng/ml). Cortisol levels were compared between multigravidae and primigravidae at each scheduled visits and presented as box and whisker plots illustration: top, bottom and middle of box, representing 25th, 75th and 50th percentiles, respectively. In both group, cortisol concentration increased in a sawtooth pattern during pregnancy with peaks at V2(28th±1 week) and V4 32th ±1 week), before decreasing at V5 which is part of post-partum.

pregnancy (primigravidae: median 108.2 ng/ml [97.6–124.5]; multigravidae 107.6 ng/ml [100.2–119.1]) until delivery (p<0.001) (see Fig 1 and S1 Table).

At delivery, peripheral blood cortisol concentration measured during labour (162.9 ng/ml [127.3–190.5]) was higher than the concentration in umbilical cord blood (143.5 ng/ml [119.2–179.6]), but the difference was not statistically significant (p = 0.158) (Table 2). However, when stratified by gravidity, cortisol level comparison showed a statistically significant difference in umbilical cord blood only (primigravidae 170.0n g/ml [148.0–187.9]; multigravidae 119.2 ng/ml [64.8–147.6]; p = 0.019) (Table 3).

**Table 2. Cortisol and prolactin concentration in peripheral blood and umbilical cord blood at delivery.**

|  | Babies | Mothers | Overall | P-value |
|---|---|---|---|---|
| Prolactin (ng/ml) [IQR] | 529.0 [386.3–712.7] | 324.6 [206.6–375.2] | 256.9 [16.1–461.9] | **0.001** |
| Cortisol (ng/ml) [IQR] | 143.5 [119.2–179.6] | 162.9 [127.3–190.5] | 124.4 [89.8–170.0] | 0.158 |

Table 2 summarize the concentration of prolactin and cortisol in babies and mothers at delivery. The difference was statistically different for prolactin but not for cortisol.

## Variation of prolactin during pregnancy

A progressive increase of prolactin concentrations was also observed from inclusion (primigravidae 88.9 ng/ml [71.5.0–126.7]; multigravidae 33.9 ng/ml [30.4–74.8]) to delivery (primigravidae 269.2ng/ml [170.0–353.7]; multigravidae 368.0 ng/ml [266.0–565.0]). The concentration of prolactin was higher in primigravidae than multigravidae at baseline (visit 1) to visit 3 (primigravidae 334.3 ng/ml [226.0–420.5]; multigravidae 251.0 ng/ml [234.0–326.0]). The concentration of prolactin decreased from delivery to 6–7 weeks after delivery for primigravidae and multigravidae but remained higher in multigravidae than primigravidae. Nevertheless, the difference was not statistically significant during the follow-up (p = 0.58) (see Fig 2 and S1 Table).

At delivery, prolactin concentration measured in umbilical cord blood (529.0n g/ml [386.3–712.7]) was significantly higher (p = 0.001) than the concentration in peripheral blood collected during labour (324.6 ng/ml [206.6–375.2]) (Table 2). However when stratified by gravidity, the differences were not statistically significant either for peripheral blood (primigravidae 269.2 ng/ml [181.2–346.9]; multigravidae 364.7 ng/ml [261.5–436.9]; p = 0.160) or for umbilical cord blood (primigravidae 543.7 [386.3–853.1]; multigravidae 510.2 ng/ml [410.2–566.4] p = 0.529) (Table 3).

## Cortisol and prolactin concentration in non-pregnant women

The cortisol and prolactin concentrations in non-pregnant women are summarised on Table 4. Nulliparous had higher cortisol (104.6 ng/ml [82.8–122.4]) and prolactin (16.7 ng/ml [15.5–25.4]) concentrations than primiparous/multiparous (cortisol 85.0 [72.7–99.5]; prolactin 9.1 [6.84–13.3]). Statistical analysis showed that the difference was significant for prolactin (p < 0.001) but not for cortisol (p = 0.095) (Table 4). The average concentration of these hormones was higher in pregnant women (cortisol 162.9ng/ml [127.3–190.5]; prolactin 324.6ng/ml [206.6–375.2]) than in non-pregnant women (cortisol 89.5ng/l [76.8–107.4]; prolactin 14.3ng/ml [8.9–16.7]) (see Figs 1 and 2, Tables 2 and 4). At week 6–7 after delivery, the concentration of prolactin remained much higher in pregnant women (205.5ng/ml [116.5–281.0]) than non-pregnant women (14.3 [8.9, 16.7]) (see Fig 2 and Table 4). However for cortisol, the concentration was higher in non-pregnant women (89.5ng/ml [76.8,– 107.4]) than in pregnant women (66.4ng/ml [47.9–96.5]) (see Fig 1 and Table 4).

**Table 3. Stratification of cortisol and prolactin concentration by gravidity in peripheral blood and umbilical blood at delivery.**

|  |  | Primigravidae | Multigravidae | Overall | P-value |
|---|---|---|---|---|---|
| Peripheral blood | Prolactin (ng/ml) [IQR] | 269.2 [181.2–346.9] | 364.7 [261.5–436.9] | 324.6 [206.6–375.2] | 0.160 |
|  | Cortisol (ng/ml) [IQR] | 172.7 [159.9–190.5] | 135.0 [119.0–181.4] | 162.9 [127.3–190.5] | 0.143 |
| Umbilical blood | Prolactin (ng/ml) [IQR] | 543.7 [386.3–853.1] | 510.2 [410.2–566.4] | 529.0 [386.3–712.7] | 0.529 |
|  | Cortisol (ng/ml) [IQR] | 170.0 [148.0–187.9] | 119.2 [64.8–147.6] | 143.5 [119.2, 179.6] | **0.019** |

Table 3 summarize the stratification of prolactin and cortisol by gravidity. The difference was only statistically different for cortisol of umbilic blood.

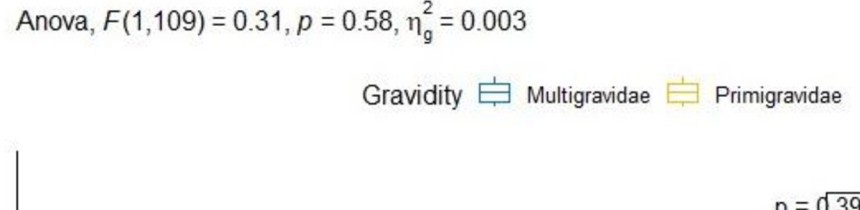

Anova, $F(1,109) = 0.31$, $p = 0.58$, $\eta_g^2 = 0.003$

**Fig 2. Dynamic of prolactin in pregnant women during pregnancy.** Legend: NS: not significant; V: visit. Code: *: <0.05; **: <0.01; ***: <0.001; ****: <0.0001. 30μl of plasma was used to determine the concentration of prolactin in blood (ng/ml). Prolactin levels were compared between multigravidae and primigravidae at each scheduled visits as described in "Method" section and presented as box and whisker plots illustration: top, bottom and middle of box, representing 25th, 75th and 50th percentiles, respectively. In multigravidae, the prolactin concentration increase from V1 (enrolment) to V4 (delivery) before decreasing at V5. For primigravidae, this concentration increase from V1 to V3 before decreasing at V4 and V5.

**Table 4. Cortisol and prolactin concentration in non-pregnant women.**

|  | Primiparous/Multiparous | Nulliparous | Overall | P-value |
|---|---|---|---|---|
| Prolactin (ng/ml) [IQR] | 9.1 [6.84, 13.3] | 16.7 [15.52, 25.4] | 14.3 [8.9, 16.7] | **<0.001** |
| Cortisol (ng/ml) [IQR] | 85.7 [72.73, 99.5] | 104.6 [82.83, 122.4] | 89.5 [76.8, 107.4] | 0.095 |

Table 4 report the concentration of cortisol and prolactin in primiparous/multiparous and nulliparous. The difference was only statistically different for prolactin.

**Table 5. Interplay or correlation of prolactin on cortisol during pregnancy.**

| Predictors | Cortisol | | |
|---|---|---|---|
| | Estimates | 95% CI | p |
| (Intercept) | 117.72 | 85.44–150.01 | <**0.001** |
| Prolactin | -0.04 | -0.08 –-0.01 | **0.009** |
| Primigravidae | 12.36 | -2.34–27.07 | 0.099 |
| Positive malaria microscopy | -3.15 | -17.94–11.65 | 0.675 |

Table 5 report the correlation of prolactin on cortisol and seems that the increase of the increase of prolactin of one unit (1.0 ng/ml), the concentration of cortisol decrease by 0.04 ng/ml.

## Interplay or variation between the cortisol and prolactin during follow-up

To assess the interplay between cortisol and prolactin during the follow-up, the variation of both hormones has been adjusted by gestation age, malaria infection and the correlation of the data have been also considered. The analysis showed that the average concentration of cortisol decreases by -0.04 ng/ml (95%CI -0.08 –-0.01) compared to the baseline concentration (117.7 ng/ml, 95%CI 85.44–150.01) when the concentration of prolactin increases by one unit (1.0 ng/ml) (p = 0.009) (Table 5). However, when the concentration of cortisol increases by one unit (1.0 ng/ml), the average concentration of prolactin decreases by -1.16 ng/ml (95%CI -2.07 –-0.25) compared to the baseline concentration (355.5 ng/ml, 95%CI 191.14–519.88) (p = 0.013) (Table 6).

The gravidity was associated to the decrease of prolactin (-3.48ng/ml, 95%CI -2.07 –-0.25) (Table 5) and the increase of cortisol (12.36ng/ml, 95%CI -2.34–27.07) in primigravidae compared to multigravidae, but the difference was not statistically significant (p = 0.099) (Table 6).

## Gestational age and malaria infection

The proportion of asymptomatic *P. falciparum* malaria at baseline (visit 1) in pregnant women was 57.70% (15/26) *versus* 11.11% (3/27) in non-pregnant women. No case of symptomatic malaria was reported during the study. The geometric median parasite was 1,628 parasite/µl (range: 111–47,802). Among pregnant women, asymptomatic malaria infection in primigravidae was two-times higher than in multigravidae (10/13 *versus* 5/15, p = 0.02) at enrolment (visit 1: 16[th]-20[th] weeks) with a geometric median parasite of 1,725 parasites/µl (range: 111–47,802) and 1,628 parasites/µl (range: 216–5,282) for primigravidae and multigravidae, respectively. In non-pregnant women, only the nulliparous were infected (3/15) with a geometric median parasite of 471 parasites/µl (range: 158–537). During the scheduled visits, the prevalence decreased among pregnant women. At visit 3 (32[th]±1 weeks of pregnancy), malaria

**Table 6. Interplay or correlation of cortisol on prolactin during pregnancy.**

| Predictors | Prolactin | | |
|---|---|---|---|
| | Estimates | 95% CI | p |
| (Intercept) | 355.51 | 191.14–519.88 | <**0.001** |
| Cortisol | -1.16 | -2.07 –-0.25 | **0.013** |
| Primigravidae | -3.48 | -80.59–73.63 | 0.929 |
| Positive malaria microscopy | 32.39 | -45.82–110.59 | 0.414 |

Table 6 report the correlation of cortisol on prolactin and seems that the increase of the increase of cortisol of one unit (1.0 ng/ml), the concentration of prolactin decrease by -3.15 ng/ml.

**Table 7. Malaria infection by gestation and gestational age.**

| | Geometric median parasite density /μl (range: lower–upper) | |
|---|---|---|
| Visits | Primigravidae | Multigravidae |
| V1 (16–20 weeks) | 1,725 (111–47,802) | 1628 (216–5,282) |
| V2 (28±1 weeks) | 5,619 (-) | 2,725 (-) |
| V3 (32±1 weeks) | 4,297 (793–7,801) | - |
| V4 (delivery) | - | 1,390 (736–2,044) |
| V5 (6–7 weeks after delivery) | 80 (-) | 56 (-) |

Table 7 reported high geometrical median parasite density in primigravidae than multigravidae.

infection was detected only in 2 primigravidae women and none in multigravidae and inversely at visit 4 (2 infected in multigravidae and none in primigravidae) (Table 7). Moreover, malaria infection concurrently leads to an increase of the average concentration of prolactin (32.4 ng/ml; 95% CI: -45.82–110.59; p = 0.414) and a decrease of cortisol (-3.2 ng/ml; 95% CI: -17 .94–11.65; p = 0.675), though these changes were not statistically significant (Tables 5 and 6).

## 4. Discussion

Our study showed a significant progressive increase of cortisol concentration, with the highest concentrations reported in primigravidae compared to multigravidae. Although this progressive increase was not significant for prolactin, a "downward regulation effect" has been reported between both hormones. These findings show that the variation of these hormones during pregnancy are related to the gestational age. Indeed, a first pregnancy is a new experience for women that could have a psychological impact on the future mother [35]. However, cortisol and prolactin are often up-regulated in a context of stress [36–38], with very different immunological functions: cortisol being an inhibitor of immune system [39–41] and prolactin a stimulator [42, 43]. The downward regulation effect that both hormones have on each other during the pregnancy when each increase to 1 unit (1.0 ng/ml) reported in this study could also explain this antagonistic immunological function of both hormones. Moreover, this regulation despite the up regulation of both hormones can also explains the limits of the concept of "suppression of immune response" during pregnancy. The increase of prolactin and cortisol levels during pregnancy with the high impact of down regulation of cortisol on prolactin compared to prolactin on cortisol, suggests that sustained increase of cortisol concentration can explain the increased susceptibility of pregnant women, particularly in primigravidae who live in malaria endemic areas, but not suppression of immune response [34] as highlighted previously. This is not the case in the present study where despite the increase in cortisol level, malaria positivity decreased. This can be due to the intermittent preventive treatment (IPTp) given to all the pregnant women during antenatal consultation. Nevertheless, the psychological impact of the first pregnancy could partly explain the higher production of cortisol and prolactin in primigravida compared to multigravida as well as the higher prevalence of malaria at enrolment. In addition to stress, which upregulates cortisol production during pregnancy, the placenta also produces corticotropin which, appears to be autonomous and not subject to the normal feedback control of glucocorticoids [44, 45]. As for prolactin, its gradual increase during pregnancy is necessary and could be explained by the body's need to maintain immune competence. Although the role of cortisol and prolactin in the reaction of the immune system has not been addressed, the graduate increase in the two hormones (with concentrations of

prolactin higher than cortisol) during pregnancy could be explained by the need for the maternal body to regulate its immune response to create a favourable environment for a successful pregnancy. Other factors such as gluconeogenesis [46, 47] and preparation of the mammary glands for breastfeeding [48] could also explain the increase in cortisol and prolactin during pregnancy and the reported differences between the two groups.

Childbirth is a time of stress, both for the mother and for the future baby. The relationship between high concentrations of cortisol and prolactin at delivery has already been reported by Bouyou-Akotet et al. in 2005 [34]. This could explain the vulnerability of primigravidae to infections such as placental malaria approaching childbirth compared to multigravidae, which represent a risk for the foetus in primigravidae. Hypercortisolaemia at childbirth is thought to be influenced by factors such as pain-related anxiety and distress [49]. Thus, vaginal delivery is the most stressful, as indicated by high levels of cortisol circulating in the peripheral blood. The decrease in prolactin at childbirth has been reported previously [50, 51]. However, the reasons for this decline are divergent. Some authors believe that it is linked to unknown factors that inhibit the release of prolactin naturally stimulated by labour stress [51], while other authors evoke the idea of a non-response of the decidual tissue to stress due to labour [50]. In our study, the reasons of the decline of prolactin only observed in primigravidae have not been assessed. However, this could be due to the psychological impact of the first pregnancy on the hypothalamic mechanisms involved in its production [35, 52, 53]. However, additional researches are needed to validate this hypothesis in primigravida during childbirth. Our study showed a significant increase of prolactin and a significant decrease of cortisol in umbilical cord blood compared to peripheral blood during labour. This may be due to the placenta/decidual tissue [54]. Indeed, placental hormones include members of prolactin and growth hormone family [55]. The high concentrations of prolactin and cortisol observed in umbilical cord blood at delivery are likely to be placental origin. Nevertheless, their role in the development of new-born immunity and the prevention to infectious disease such as malaria during the first year of life remains unclear.

After childbirth and the expulsion of the placenta, the cortisol level returns to the pre-pregnancy level, while the prolactin level remains high. The decrease in cortisol is due to the absence of the placenta and the decrease of stress-associated childbirth, which are the main source of corticotrophin under the action of a permanent positive feedback and the resumption of control by the hypothalamic-pituitary axis. The high levels of prolactin could be explained by the sucking of the baby during breastfeeding, which then becomes the main stimulus that induces the secretion of prolactin by the lactotroph cells of the anterior pituitary after birth [56]. The high postpartum prolactin level promotes the stabilization and maintenance of milk synthesis [57]. Lactational hyperprolactinemia also prevents early postpartum pregnancy.

This is an explorative study conducted in pregnant women. However, this did not affect the findings of the present study, but allowed to lay the basis for future investigation of the roles of these endocrine hormones in immune system during pregnancy. A possible limitation is the enrolment period of pregnant and non-pregnant women. This could affect the expected results given the fact that malaria transmission in the study area was seasonal.

## 5. Conclusion

These findings showed that the concentrations of cortisol and prolactin throughout pregnancy were up-regulated by gestation except for primigravidae pregnant women for whom this hormone decreases at delivery and cortisol between the visits 2 and 3. A downward regulation effect between the two hormones was also reported, which seems to be important for both foetus and mother.

## Supporting information

**S1 Checklist. STROBE statement—Checklist of items that should be included in reports of** *cohort studies.*
(DOCX)

**S1 Table. Concentration of cortisol and prolactin at scheduled visit and delivered.** This table summarize the value of cortisol and prolactin at scheduled visit.
(DOCX)

**S1 Data.**
(XLSX)

## Acknowledgments

We would like to thank the study staff of the rural health facilities of Nanoro, Seguedin and Nazoanga in the Health District of Nanoro for their valuable contributions to the work. We are indebted to the pregnant and non-pregnant women for their participation in the study.

## Author Contributions

**Conceptualization:** Francois Kiemde, Hermann Sorgho, Berenger Kabore, Halidou Tinto.

**Data curation:** Serge Henri Zango, Ousmane Traore, Berenger Kabore.

**Formal analysis:** Francois Kiemde, Toussaint Rouamba, Ousmane Traore, Berenger Kabore, Hamtandi Magloire Natama.

**Funding acquisition:** Francois Kiemde, Innocent Valea, Halidou Tinto.

**Investigation:** Francois Kiemde, Serge Henri Zango, Gnohion Fabrice Some, Berenger Kabore, Henk Schallig, Halidou Tinto.

**Methodology:** Francois Kiemde.

**Project administration:** Francois Kiemde, Hermann Sorgho, Serge Henri Zango, Gnohion Fabrice Some, Halidou Tinto.

**Resources:** Halidou Tinto.

**Software:** Toussaint Rouamba.

**Supervision:** Francois Kiemde, Hermann Sorgho, Gnohion Fabrice Some, Yeri Esther Hien, Halidou Tinto.

**Validation:** Francois Kiemde, Hamtandi Magloire Natama.

**Writing – original draft:** Francois Kiemde, Hermann Sorgho, Serge Henri Zango, Gnohion Fabrice Some, Toussaint Rouamba, Ousmane Traore, Berenger Kabore, Hamtandi Magloire Natama, Yeri Esther Hien, Innocent Valea, Henk Schallig, Halidou Tinto.

**Writing – review & editing:** Francois Kiemde, Hermann Sorgho, Serge Henri Zango, Gnohion Fabrice Some, Toussaint Rouamba, Ousmane Traore, Berenger Kabore, Hamtandi Magloire Natama, Yeri Esther Hien, Innocent Valea, Henk Schallig, Halidou Tinto.

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
