## [Decision Letter · Decision Letter 0]

31 Jan 2024

PONE-D-23-33355Effects of gestational age on blood cortisol and prolactin levels during pregnancy in malaria endemic areaPLOS ONE

Dear Dr. Kiemde,

Thank you for submitting your manuscript to PLOS ONE. After careful consideration, we feel that it has merit but does not fully meet PLOS ONE’s publication criteria as it currently stands. Therefore, we invite you to submit a revised version of the manuscript that addresses the points raised during the review process.

We look forward to receiving your revised manuscript.

Kind regards,

Ibrahim Sebutu Bello, MBBS, MPH, MD, FMCGP

Academic Editor

PLOS ONE

Journal Requirements:

3. Thank you for stating the following financial disclosure: "International Society for Infectious Diseases (ISID), Small Grant 2014, “Determination of cytotoxicity of NK cells against erythrocytes infected by Plasmodium Falciparum during pregnancy”."

4. We note that your Data Availability Statement is currently as follows: All relevant data are within the paper.

6. We notice that your supplementary tables are included in the manuscript file. Please remove them and upload them with the file type 'Supporting Information'. Please ensure that each Supporting Information file has a legend listed in the manuscript after the references list.

Reviewers' comments:

Reviewer's Responses to Questions

**Comments to the Author**

1. Is the manuscript technically sound, and do the data support the conclusions?

Reviewer #1: Partly

Reviewer #2: Partly

2. Has the statistical analysis been performed appropriately and rigorously? 

Reviewer #1: No

Reviewer #2: I Don't Know

3. Have the authors made all data underlying the findings in their manuscript fully available?

Reviewer #1: Yes

Reviewer #2: Yes

4. Is the manuscript presented in an intelligible fashion and written in standard English?

Reviewer #1: Yes

Reviewer #2: Yes

5. Review Comments to the Author

Reviewer #1: The manuscript investigates the effect of gestation on cortisol and prolactin levels in pregnant women, comparing primigravidae and multigravidae. The study design, methodology, and data analysis are well-described, and the results are presented clearly and concisely. The discussion adequately interprets the findings in the context of existing literature.

Major Comments:

1. Background:

o The introduction could be strengthened by providing a more focused review of the role of cortisol and prolactin in pregnant women, particularly in relation to immune function and stress response.

o The concept of "suppression of (some) immune responses" could be further elaborated upon, differentiating between specific types of immune responses and the nuances involved.

2. Methods:

o The inclusion and exclusion criteria for participants should be clearly stated, especially for the control groups.

o It would be helpful to explain the rationale behind the chosen time points for sample collection.

o The description of the laboratory procedures could be expanded to include details such as the manufacturers and specific kits used for the cortisol and prolactin assays.

3. Results:

o The tables and figures could be improved by adding more descriptive captions and legends.

o Consider including statistical tests and p-values in the figures directly.

o It would be helpful to present the results of the correlation analysis between cortisol and prolactin in a separate table.

4. Discussion:

o The discussion could be strengthened by addressing limitations of the study, such as the relatively small sample size and potential confounding factors.

o Consider including more references to support the claims made about the roles of cortisol and prolactin in pregnancy.

Minor Comments:

• Consider using a consistent formatting style for references throughout the manuscript.

• Ensure that all abbreviations are defined upon first use.

• Check for typos and grammatical errors.

Specific Points:

• Page 5: Define IPT (immunologic pregnancy test).

• Page 6: Explain how the detection limit of 100 ng/ml for prolactin was adjusted for higher concentrations.

• Page 7: Clarify what is meant by "non-response" in the sample size calculation.

• Page 7: Mention the statistical test used for comparing cortisol and prolactin concentrations between gravidity groups.

• Page 7: Specify the factors included in the LMER model for assessing the interplay between cortisol and prolactin.

Overall, the manuscript presents valuable data on the modulation of cortisol and prolactin levels during pregnancy. Addressing the major and minor comments will further strengthen the manuscript and improve its clarity and impact.

Reviewer #2: REVIEW COMMENTS

TITLE: Effects of gestational age on blood cortisol and prolactin levels during pregnancy in malaria endemic area

The study investigated a rather interesting and pertinent topic and the manuscript was generally well written. The specific issues that need to be addressed to improve the report are highlighted below. There are also some grammatical errors here and there that should be corrected eg. ”Primigravidae showed a higher increase of cortisol than multigravidae during pregnancy with statistically significant different(ce)”

Background

1. The first two paragraphs of the background do not appear to be relevant to the present manuscript but rather to the overall study. The background can start with the 3rd paragraph and then expanded a little bit.

Methods

2. The inclusion and exclusion criteria should be more explicit.

3. Were those with twin gestation excluded? Were women with malaria symptoms excluded, or there was no case of symptomatic malaria among those with positive tests? Only cases of asymptomatic malaria infection were mentioned in the result. This may need to be clarified. This information is more imperative considering that “Malaria, caused by Plasmodium falciparum, remains the first cause of medical consultation in pregnant women in this region…”

4. Were there any loss to follow-up or incomplete data??? If all the patients enrolled at baseline ended up completing the study. It is good to know what measures were made that ensured all respondents attended all follow-up visits, delivered at the facilities, and attended PNC.

5. Where and how were the controls recruited?

6. Sample size: the ideal way to adjust for 10% attrition is: Sample size / (1 – attrition rate)

7. Data analysis: a bit more detail would be in order. For example, which statistical test was used for the difference in quantitative variables between groups eg. hormone concentrations at delivery between babies and mothers (table 2), and between primigravidae and multigravidae (table 3), etc.

Results

8. Mode of delivery: was the mode of delivery the same (vaginal?) for all pregnant women? Did authors consider the possibility of the effect of the mode of delivery on the hormone concentration, especially cortisol, at V4-delivery?

9. Variation of cortisol during pregnancy: concerning the p-value of 0.024 in this sentence “Primigravidae showed a higher increase of cortisol than multigravidae during pregnancy with statistically significant different (p=0.024)” and also in figure 2, is it speaking to the difference in the increase between the two groups or the difference in the repeated measures across different visits for ALL pregnant respondents?

10. Variation of cortisol during pregnancy: “A gradual increase in cortisol concentrations from the enrolment to delivery was observed …with the highest concentrations measured at delivery…” This statement did not reflect the fact that there was no increase, but a decrease btw V2 and V3.

11. Variation of prolactin during pregnancy: Authors stated: “Nevertheless, the difference was not statistically significant during the follow-up (p=0.6).” Is this referring to the entire 6 visits? If that’s so, how does that juxtapose with the next statement starting with: “The (increase in??) concentration of prolactin was statistically significant from visit…”. If the latter was based on post-hoc analyses, it should be stated in the data analysis section, along with the specific statistical test used.

12. Figures 2&3: There is an indication that an ANOVA test was used for the analyses. Such information should be included in the data analysis section. However, considering the way the cortisol and prolactin concentrations were summarized and presented using the median, it suggests that they were NOT normally distributed (which can also be deduced from the box plots), hence a non-parametric alternative to repeated measure ANOVA may be more appropriate.

13. In all relevant tables, I suggest that Q1-Q3 be replaced with IQR, to synchronise with the way it was represented in the main text

14. Table 3: Some p-values are incorrectly written, have commas

15. Table 3: Kindly crosscheck the title, seems should be “Stratification….by gravidity….”

16. In all relevant tables, I suggest that Q1-Q3 be replaced with IQR, to synchronise with the way it was represented in the main text

17. Table 3: Some p-values are incorrectly written, have commas

18. Table 3: Kindly crosscheck the title, seems should be “Stratification….by gravidity….”

Discussion

19. In the study, while cortisol increased with GA, malaria positivity decreased. Hence, the following statement does not seem to be completely supported by the study findings: “The increase of prolactin and cortisol levels during pregnancy with the high impact of down regulation of cortisol on prolactin compared to prolactin on cortisol suggests that sustained increase of cortisol concentration can explain the increased susceptibility of pregnant women, particularly in primigravidae who live in malaria endemic areas”.

20. Conclusion: This needs to be revised for the following reasons:

a) The statement “…except for primigravidae pregnant women for whom this hormone decreases at delivery” is not true for cortisol”.

b) I have concerns about the conclusion of a ‘negative correlation’ based on the interpretation of the results. Negative correlation suggests that they were moving in the opposite direction i.e. while one was increasing, the other was decreasing. It seems to me that the interpretation should rather be that the rate or pattern of increase was different for both hormones i.e. the rate was higher for one than for the other, because to a large extent, both increased from baseline.

6. PLOS authors have the option to publish the peer review history of their article (what does this mean?). If published, this will include your full peer review and any attached files.

Reviewer #1: **Yes: **Abhilasha Sharma

Reviewer #2: No

---

## [Author Response · Author response to Decision Letter 0]

25 Mar 2024

REBUTTAL TO THE POINTS RAISED BY Editor:

Comment 1: Please ensure that your manuscript meets PLOS ONE's style requirements, including those for file naming. The PLOS ONE style templates can be found at https://journals.plos.org/plosone/s/file?id=wjVg/PLOSOne_formatting_sample_main_body.pdf and 

RESPONSE: We would like to thank the editor for this observation. Now we are made the changes to meet the PLOS ONE’s style requirement.

Comment 2: Note from Emily Chenette, Editor in Chief of PLOS ONE, and Iain Hrynaszkiewicz, Director of Open Research Solutions at PLOS: Did you know that depositing data in a repository is associated with up to a 25% citation advantage (https://doi.org/10.1371/journal.pone.0230416)? If you’ve not already done so, consider depositing your raw data in a repository to ensure your work is read, appreciated and cited by the largest possible audience. You’ll also earn an Accessible Data icon on your published paper if you deposit your data in any participating repository (https://plos.org/open-science/open-data/#accessible-data).

RESPONSE: Thank to the editor to remind us about the advantage of depository data in our research career. We are planned to deposit this database on repository as soon as the main data are published.

Comment 3: Thank you for stating the following financial disclosure: "International Society for Infectious Diseases (ISID), Small Grant 2014, “Determination of cytotoxicity of NK cells against erythrocytes infected by Plasmodium Falciparum during pregnancy”."

RESPONSE: Now we mentioned the role of the funder in the project in the cover letter above. Please note that the funder had no role in the study design, data collection and analysis, decision to publish, or preparation of the manuscript.

Comment 4: We note that your Data Availability Statement is currently as follows: All relevant data are within the paper.

RESPONSE: Thank to the editor to pointed this out. As mentioned above, all relevant data are within the paper. However, the data that support the findings of this study are available on request from the corresponding author, FK. The data are not publicly available due as their containing information that could compromise the privacy of research participants. Please you can go ahead and change the online submission form on my behalf.

Comment 5: We note that Figure 1 in your submission contain [map/satellite] images which may be copyrighted. All PLOS content is published under the Creative Commons Attribution License (CC BY 4.0), which means that the manuscript, images, and Supporting Information files will be freely available online, and any third party is permitted to access, download, copy, distribute, and use these materials in any way, even commercially, with proper attribution. For these reasons, we cannot publish previously copyrighted maps or satellite images created using proprietary data, such as Google software (Google Maps, Street View, and Earth). For more information, see our copyright guidelines: http://journals.plos.org/plosone/s/licenses-and-copyright.

RESPONSE: The figure 1 has been deleted to the manuscript and the manuscript has been updated. Thank.

Comment 6: We notice that your supplementary tables are included in the manuscript file. Please remove them and upload them with the file type 'Supporting Information'. Please ensure that each Supporting Information file has a legend listed in the manuscript after the references list..

RESPONSE: The supplement table has been remove to the manuscript and uploaded as “Supporting information”. It has been also renamed as “Supporting table”.

REBUTTAL TO THE POINTS RAISED BY REVIEWERS:

Reviewer #1

Comment 1: The manuscript investigates the effect of gestation on cortisol and prolactin levels in pregnant women, comparing primigravidae and multigravidae. The study design, methodology, and data analysis are well-described, and the results are presented clearly and concisely. The discussion adequately interprets the findings in the context of existing literature.

RESPONSE: We would like to thank the reviewer for his/her positive appreciation of the manuscript.

Major comments

1. Background

Comment 2: The introduction could be strengthened by providing a more focused review of the role of cortisol and prolactin in pregnant women, particularly in relation to immune function and stress response.

RESPONSE: We have now strengthened the introduction by providing a more focus review in the role: line 115 – 117: “This is the case of stress responses on immune system during pregnancy. Indeed, the immune stress can be bidirectional by enhancing or suppress immune function [21,22]”.

Comment 3: The concept of "suppression of (some) immune responses" could be further elaborated upon, differentiating between specific types of immune responses and the nuances involved.

RESPONSE: We have now reviewed the concept of “suppression of (some) immune response” and updated the references list: line 100 – 102: “Studies consider immunosuppression as a state in which immune system does not work as well it should. This mechanism provided to mother an effective protection from harmful pathogens while allowing tolerance to foetal antigens [13,14]”.

Methods

Comment 4: The inclusion and exclusion criteria for participants should be clearly stated, especially for the control groups.

RESPONSE: We have now reviewed the inclusion criteria for the control arm: line 177 – 183: “For the control groups, participants were recruitment within the catchment area of the recruitment health facilities listed above. In this group, participants included were non-pregnant nulliparous and both primiparous/ multiparous individuals. The number of nulliparous and primiparous/multiparous to be recruited by health facility catchment area were paired respectively to the number of primigravidae and multigravidae women recruited in this health facility catchment area. For each participant enrolled in the control group, venous blood was drawn once into EDTA tubes”.

Comment 5: It would be helpful to explain the rationale behind the chosen time points for sample collection.

RESPONSE: We have now explained the rationale behind the chosen time points for the sample collection: line 169 – 173: “Regarding the rationale behind these time points, the choice of 16th to 20th weeks of pregnancy is related to the fact that pregnant women consult to their first antenatal consultation at this period, due to socio-economic factors. This also represent the end of organogenesis. The choices of 28th�1 weeks of pregnancy is related to the maturation of the different organs of the foetus and 32th �1 to the pre-delivery”.

Comment 6: The description of the laboratory procedures could be expanded to include details such as the manufacturers and specific kits used for the cortisol and prolactin assays.

RESPONSE: With respect to the review related to this comment, we would like to refer to line 218 – 228: “Plasma concentrations of cortisol and prolactin were determined by using the test device ichromaTM cortisol for the quantification of cortisol and ichromaTM prolactin for the quantification of prolactin on the devices i-chamber (Weldon Biotech India Pvt. Ltd. India) and ichroma II (Boditech Med Inc. Korea) according to the procedure described by the manufacturer. For the quantification of the cortisol, 30µl of plasma were transferred to the tube containing the detection buffer provided by the manufacturer and mixed 10 times manually by pipetting. A volume of 75µl of the mixture were placed in the well of the ichromaTM cortisol cartridge and immediately incubated at 25°C for 10 minutes in the i-chamber. After incubation, the cartridge was removed and read immediately in the ichroma II. The analyzer automatically displayed the test result on the display screen, printed and validated. The procedure of quantification of prolactin was similar to cortisol, except the volume of plasma pipetted (75µl instead of 30µl)”. The details on manufacturers and cortisol and prolactin kits were included.

Results

Comment 7: The tables and figures could be improved by adding more descriptive captions and legends.

RESPONSE: We have now added descriptive captions and legends for figures and tables.

Comment 8: Consider including statistical tests and p-values in the figures directly.

RESPONSE: We would like to thank the reviewer for this observation. We have now added p-values in the figure directly.

Comment 9: It would be helpful to present the results of the correlation analysis between cortisol and prolactin in a separate table.

RESPONSE: We now present the results of the correlation analysis between cortisol and prolactin in a separate table (see table 5 and 6) and the numbers of the tables has been updated.

Discussion

Comment 10: The discussion could be strengthened by addressing limitations of the study, such as the relatively small sample size and potential confounding factors.

RESPONSE: We would like to thank the reviewer for pointed-out the sample size. This has been already highlighted in the manuscript: line 477 – 482 : “This is an explorative study conducted in pregnant women. The study has been designed to determine the cytotoxicity of NK cells against erythrocytes infected by Plasmodium falciparum. However, this did not affect the findings of the present study, but allowed to lay the basis for future investigation of the roles of these endocrine hormones in immune system during pregnancy. A possible limitation is the enrolment period of pregnant and non-pregnant women. This could affect the expected results given the fact that malaria transmission in the study area was seasonal”. This cannot be reported as a limit given the fact that this was an explorative study and the findings need to be confirm with an important sample size.

Comment 11: Consider including more references to support the claims made about the roles of cortisol and prolactin in pregnancy.

RESPONSE: We have now including more references to support the claims made about the role of cortisol and prolactin.

Minor Comments:

Comment 12: Consider using a consistent formatting style for references throughout the manuscript.

RESPONSE: We have now using consistent formatting style for references. Thank.

Comment 13: Ensure that all abbreviations are defined upon first use.

RESPONSE: We would like to thank reviewer for this observation. We have now made correction and added abbreviation paragraph to the manuscript (see line 502 – 505).

Comment 14: Check for typos and grammatical errors.

RESPONSE: These are now checked.

Specific Points:

Comment 15: Page 5: Define IPT (immunologic pregnancy test).

RESPONSE: With respect to the review related to this comment, IPT was define: line 185.

Comment 16: Page 6: Explain how the detection limit of 100 ng/ml for prolactin was adjusted for higher concentrations.

RESPONSE: With respect to the review related to this comment, we would like to refer to line 229 – 234: “The detection limit of prolactin with ichroma II is 100ng/ml. In case of concentration over 100ng/ml, ¼ and 1�8 dilution of the plasma was done with saline phosphate buffer as recommended by the manufacturer. The exact value of the prolactin was then obtained by multiplying the measured value by the dilution factor”.

Comment 17: Page 7: Clarify what is meant by "non-response" in the sample size calculation.

RESPONSE: We have now clarified: Line 266: “By considering 10% o

---

## [Decision Letter · Decision Letter 1]

24 May 2024

PONE-D-23-33355R1Effects of gestational age on blood cortisol and prolactin levels during pregnancy in malaria endemic areaPLOS ONE

Dear Dr. Kiemde,

Thank you for submitting your manuscript to PLOS ONE. After careful consideration, we feel that it has merit but does not fully meet PLOS ONE’s publication criteria as it currently stands. Therefore, we invite you to submit a revised version of the manuscript that addresses the points raised during the review process.

We look forward to receiving your revised manuscript.

Kind regards,

Ibrahim Sebutu Bello, MBBS, MPH, MD, FMCGP

Academic Editor

PLOS ONE

Journal Requirements:

Additional Editor Comments:

Kindly provide a response to all the issues raised by the reviewer

Reviewers' comments:

Reviewer's Responses to Questions

**Comments to the Author**

1. If the authors have adequately addressed your comments raised in a previous round of review and you feel that this manuscript is now acceptable for publication, you may indicate that here to bypass the “Comments to the Author” section, enter your conflict of interest statement in the “Confidential to Editor” section, and submit your "Accept" recommendation.

Reviewer #2: All comments have been addressed

Reviewer #3: All comments have been addressed

2. Is the manuscript technically sound, and do the data support the conclusions?

Reviewer #2: Partly

Reviewer #3: Yes

3. Has the statistical analysis been performed appropriately and rigorously? 

Reviewer #2: N/A

Reviewer #3: Yes

4. Have the authors made all data underlying the findings in their manuscript fully available?

Reviewer #2: Yes

Reviewer #3: Yes

5. Is the manuscript presented in an intelligible fashion and written in standard English?

Reviewer #2: Yes

Reviewer #3: Yes

6. Review Comments to the Author

Reviewer #2: (No Response)

Reviewer #3: Dear authors

Many thanks for your research

Here are some comments on your original research.

1. In line 208 and 209 “The detection limit of prolactin with ichroma II is 100ng/ml. In case of concentration over 100ng/ml, ¼ and 1�8 dilution of the plasma was done with saline phosphate buffer as recommended” the detection limit is an incorrect phrase. In clinical laboratory, we describe the detection limit as: the lowest concentration that the procedure (laboratory kit) can detect. The highest concentration that the procedure or kit can detect and higher concentrations should be diluted is defined as the linearity range. Therefore, dear authors should correct this definition. Please, refer to kit inset sheet/instruction about the linearity and detection limit complementary descriptions.

2. In figures 1 and 2 (box plots), please, correct the p-values decimals. Please, remove extra decimals.

3. In discussion section, I think “the first year of live” is better to be replaced with “the first year of life”.

4. In discussion section, Lines 411 and 412 “This is an explorative study conducted in pregnant women. The study has been designed to determine the cytotoxicity of NK cells against erythrocytes infected by Plasmodium falciparum.” This phrase is related to which study? If it is related to the current study, I can not see any direct variable related to NK cells and immune system. If this is not related to the current study, please remove it from discussion and abstract and elsewhere in body text.

Many thanks with the best wishes

7. PLOS authors have the option to publish the peer review history of their article (what does this mean?). If published, this will include your full peer review and any attached files.

Reviewer #2: No

Reviewer #3: **Yes: **Abdorrahim Absalan

---

## [Author Response · Author response to Decision Letter 1]

4 Jul 2024

REBUTTAL TO THE POINTS RAISED BY REVIEWERS:

Reviewer #2 (No response)

Reviewer #3

General comment: Dear authors, Many thanks for your research. Here are some comments on your original research.

RESPONSE: We would like to thank the reviewer for his/her time to review the resubmission of the manuscript. All the comments have been addressed in the review manuscript.

Comment 1: In line 208 and 209 “The detection limit of prolactin with ichroma II is 100ng/ml. In case of concentration over 100ng/ml, ¼ and 1�8 dilution of the plasma was done with saline phosphate buffer as recommended” the detection limit is an incorrect phrase. In clinical laboratory, we describe the detection limit as: the lowest concentration that the procedure (laboratory kit) can detect. The highest concentration that the procedure or kit can detect and higher concentrations should be diluted is defined as the linearity range. Therefore, dear authors should correct this definition. Please, refer to kit inset sheet/instruction about the linearity and detection limit complementary descriptions.

RESPONSE: We would like to thank the reviewer for this observation regarding the detection limit. We have now corrected the manuscript: line 208: “The threshold for prolactin quantification with ichroma II is 100ng/ml”.

Comment 2: In figures 1 and 2 (box plots), please, correct the p-values decimals. Please, remove extra decimals

RESPONSE: We would like to thank the reviewer for this observation. We have now arranged the p-values in the figure directly and add a legend to the figure.

Comment 3: In discussion section, I think “the first year of live” is better to be replaced with “the first year of life”.

RESPONSE: Now we are corrected. Please see line 401 – 403: “Nevertheless, their role in the development of new-born immunity and the prevention to infectious disease such as malaria during the first year of life remains unclear”. 

Comment 4: In discussion section, Lines 411 and 412 “This is an explorative study conducted in pregnant women. The study has been designed to determine the cytotoxicity of NK cells against erythrocytes infected by Plasmodium falciparum.” This phrase is related to which study? If it is related to the current study, I can not see any direct variable related to NK cells and immune system. If this is not related to the current study, please remove it from discussion and abstract and elsewhere in body text. Many thanks with the best wishes

RESPONSE: We have now deleted the following sentence to the manuscript: “The study has been designed to determine the cytotoxicity of NK cells against erythrocytes infected by Plasmodium falciparum”.

---

## [Decision Letter · Decision Letter 2]

30 Aug 2024

Effects of gestational age on blood cortisol and prolactin levels during pregnancy in malaria endemic area

PONE-D-23-33355R2

Dear Dr. Kiemde,

We’re pleased to inform you that your manuscript has been judged scientifically suitable for publication and will be formally accepted for publication once it meets all outstanding technical requirements.

Kind regards,

Ibrahim Sebutu Bello, MBBS, MPH, MD, FMCGP

Academic Editor

PLOS ONE

Reviewers' comments:

Reviewer's Responses to Questions

**Comments to the Author**

1. If the authors have adequately addressed your comments raised in a previous round of review and you feel that this manuscript is now acceptable for publication, you may indicate that here to bypass the “Comments to the Author” section, enter your conflict of interest statement in the “Confidential to Editor” section, and submit your "Accept" recommendation.

Reviewer #2: All comments have been addressed

2. Is the manuscript technically sound, and do the data support the conclusions?

Reviewer #2: (No Response)

3. Has the statistical analysis been performed appropriately and rigorously? 

Reviewer #2: (No Response)

4. Have the authors made all data underlying the findings in their manuscript fully available?

Reviewer #2: (No Response)

5. Is the manuscript presented in an intelligible fashion and written in standard English?

Reviewer #2: (No Response)

6. Review Comments to the Author

Reviewer #2: (No Response)

7. PLOS authors have the option to publish the peer review history of their article (what does this mean?). If published, this will include your full peer review and any attached files.

Reviewer #2: No

---

## [Editor Report · Acceptance letter]

3 Sep 2024

PONE-D-23-33355R2 

PLOS ONE

Dear Dr. Kiemde, 

I'm pleased to inform you that your manuscript has been deemed suitable for publication in PLOS ONE. Congratulations! Your manuscript is now being handed over to our production team.

Kind regards, 

on behalf of

Dr. Ibrahim Sebutu Bello 

Academic Editor

PLOS ONE